# COVID-19: Why Do People Refuse Vaccination? The Role of Social Identities and Conspiracy Beliefs: Evidence from Nationwide Samples of Polish Adults

**DOI:** 10.3390/vaccines10020268

**Published:** 2022-02-10

**Authors:** Marta Marchlewska, Katarzyna Hamer, Maria Baran, Paulina Górska, Krzysztof Kaniasty

**Affiliations:** 1Institute of Psychology, Polish Academy of Sciences, 1 Jaracza St., 00-378 Warsaw, Poland; khamer@psych.pan.pl (K.H.); kaniasty@iup.edu (K.K.); 2Faculty of Psychology, SWPS University of Social Sciences and Humanities, 19/31 Chodakowska St., 03-815 Warsaw, Poland; mbaran@swps.edu.pl; 3Faculty of Psychology, University of Warsaw, 5/6 Stawki St., 00-183 Warsaw, Poland; paulina.gorska@psych.uw.edu.pl; 4Department of Psychology, Indiana University of Pennsylvania, Indiana, PA 15701, USA

**Keywords:** willingness to vaccinate against COVID-19, COVID-19 conspiracy beliefs, national narcissism, national identification, identification with all humanity

## Abstract

In the present research, we focus on COVID-19 vaccine hesitancy, and empirically examine how different forms of social identity (defensive vs. secure national identity and identification with all humanity) and conspiracy beliefs are associated with COVID-19 vaccine hesitancy. In two cross-sectional nationwide surveys (Study 1, *n* = 432, and Study 2, *n* = 807), we found that willingness to vaccinate against COVID-19 was negatively linked to national narcissism, but positively related to a secure national identification, that is, national identification without the narcissistic component. In both studies, we also found that the relationship between narcissistic (vs. secure) national identity and unwillingness to vaccinate against COVID-19 was mediated by COVID-19 vaccine conspiracy beliefs. These effects were present even when we accounted for basic demographics (Studies 1 and 2) and identification with all humanity (Study 2), which had been found to be a significant predictor of health behaviors during COVID-19. In line with previous research, identification with all humanity was positively associated with the willingness to vaccinate against COVID-19. We discuss the implications for understanding the role of the way in which people identify with their national and supranational groups in antiscience attitudes and (mal)adaptive behaviors during COVID-19 pandemic.

## 1. Introduction

So far during the pandemic, various factors have impacted whether new COVID-19 cases globally increase or decline in countries. Among these factors are infection prevention policies, mutations of the coronavirus itself, and human responses to the COVID-19 pandemic, such as practicing physical distancing, handwashing, mask wearing, and especially attitudes toward the COVID-19 vaccines [1,2]. Although the overwhelming scientific consensus is clear regarding vaccine safety and their potential to significantly weaken the link between cases, serious illness, and death, many people choose to not be vaccinated [3]. Why does it happen, and how can we address this problem? To answer these questions scholars and practitioners investigate not only social and political factors, but also psychological characteristics that could predict and thereby ultimately prevent COVID-19 vaccine hesitancy. For example, previous research found that low trust in vaccines in general [4], concerns about side effects related to vaccines [5], distrust in government or healthcare professionals [6], the need for closure [7], and conspiracy beliefs contributed to factors that generate more doubts and objections to COVID-19 vaccination [8,9]. In this work, we focus on the latter aforementioned factors and analyze the role of social identities in seizing on COVID-19 vaccine conspiracy beliefs that, according to our assumptions, would further lead to lower willingness to vaccinate against COVID-19.

### 1.1. COVID-19 Vaccine Conspiracy Beliefs—Causes and Consequences

Shortly after the European Union had officially launched its program of mass vaccination against COVID-19 in December 2020, conspiracy theories and misconceptions about COVID-19 vaccines began to circulate online [10,11,12,13]. These theories often accuse scientists, governmental agencies, and pharmaceutical companies of concealing information from the public about the dangers associated with vaccination, which is one of the reasons for a decrease in intention to vaccinate. Conspiracy theories frequently claim that COVID-19 vaccines contain microchips that track the location of patients, are generally harmful, and result in serious diseases, all of which is covered up by powerful and malevolent groups [14]. Importantly, COVID-19 vaccine conspiracy beliefs, just as the majority of similar convictions [15,16], involve an intergroup dimension. They postulate that there is a nefarious outgroup (e.g., scientists and Big Pharma) that covertly influences the course of COVID-19 pandemic and thus constitutes a threat to the ingroup (e.g., one’s own nation).

This intergroup facet of conspiracy theories has inspired researchers to analyze conspiracy beliefs in relation to social and political identities. For example, many research projects [17,18,19] emphasized the role of narcissistic group identity in seizing on conspiratorial explanations. This is not surprising, as collective narcissism (i.e., a belief in one’s group’s greatness that is contingent on external validation [20]) increases sensitivity to intergroup threats [16], making people distrustful and more prone to interpret even ambivalent intergroup situations as posing a real danger to their own group [21]. This usually leads to an appraisal of the world as a dangerous and nefarious place, and a search for imaginary enemies (i.e., alleged conspirators) that could be blamed for the ingroup’s misfortunes [19]. In line with this logic, Cislak and colleagues [22] found that defensiveness about one’s nationality, operationalized as national narcissism, predicted vaccination conspiracy beliefs, which further led to lower support for a voluntary vaccination policy. In such a way, due to exaggerating feelings of intergroup threat coming from higher-status groups (e.g., scientists or pharmaceutical companies), national narcissists were found to perceive the world as a dangerous place, seize on conspiracy beliefs, and in turn score higher on vaccine hesitancy. Although inspiring, the Cislak et al. [22] study did not assess the actual willingness to vaccinate against COVID-19, which is crucial in the situation when the world is facing the pandemic, a global crisis with millions of deaths.

We aimed to fill this gap and better understand the role of social identities (i.e., national narcissism, secure national identification, identification with all humanity) and COVID-19 vaccine conspiracy beliefs in predicting attitudes towards COVID-19 vaccines. Specifically, we sought to examine which forms of social identity may be linked to COVID-19 conspiracy beliefs and the unwillingness to vaccinate against COVID-19. We investigated whether those who score high (vs. low) on national narcissism would be more (vs. less) prone to endorse COVID-19 vaccine conspiracy theories and, in turn, express less (vs. more) willingness to vaccinate against COVID-19. We also controlled for a secure national identification (i.e., national identity free from national narcissism; Studies 1 and 2) and identification with all humanity (Study 2) to check whether the effects of national narcissism on vaccine hesitancy would be present after accounting for these variables previously found to predict health behaviors in the context of COVID-19 [23,24]. In such a way, we offer the first empirical insight into associations between different forms of social identities, COVID-19 vaccine conspiracy beliefs, and the (un)willingness to vaccinate against COVID-19.

### 1.2. Social Identities—Different Forms and Their Correlates

Addressing people’s beliefs about their ingroups is becoming increasingly important in the general quest for understanding what is going on in the world today [25] and in the context of mass emergencies [26], particularly including the COVID-19 the pandemic [27]. For example, our previous research found that the way people perceive their nation can shape not only their political preferences (e.g., voting [28]), but also existential choices (e.g., in which country they want to live [29]). These dynamics depend, however, on whether individuals’ attachment to a nation is defensive (i.e., stems from frustrated individual or collective needs) or secure (i.e., is built on the foundations of individual or collective satisfaction). The present investigation focuses on three different types of identities: national narcissism (Studies 1 and 2), secure national identification (Studies 1 and 2), and identification with all humanity (Study 2). The national narcissism type of identity is defensive and can be related to maladaptive psychological outcomes at both the individual and group levels. Secure national identification and identification with all humanity are affirmative identities that can be linked to positive psychological outcomes [30,31].

National narcissism is characterized by a grandiose appraisal of one’s national group that is contingent on the external acknowledgement of its worth [20,32,33]. This type of ingroup identity is compensatory and destructive from the perspective of both individual and group-level processes [20,30,34,35]. Collective narcissists search for a strong group that would help them manage some of their psychological shortcomings (group in the service of the self [36]). For example, previous studies showed that narcissistic identification stems from a sense of low personal control [37] and low self-esteem [38]. Collective narcissism is associated with high defensive self-evaluation [39] and high anxious attachment style [36]. Although national narcissism results from the frustration of different needs, it does not necessarily reinforce feelings of personal control or boost self-esteem [37]. In contrast, due to its defensive nature, it backfires, leading to maladaptive psychological outcomes. For example, it was found to predict negative view of human nature (i.e., social cynicism [39]), conspiracy beliefs [16], ingroup disloyalty [29], intergroup hostility [40,41], and lower intergroup forgiveness [34]. Bertin and colleagues [17] reported that national narcissism was associated with the rejection of climate science, an effect that was mediated by climate change conspiracy beliefs. Thus, there are reasons to believe that the distrustful nature of collective narcissists would make them more prone to endorse COVID-19 vaccine conspiracy beliefs and subsequently lead to lower readiness to vaccinate against COVID-19.

We would expect very different dynamics among secure identifiers whose national commitment stems from satisfied psychological needs (e.g., higher feelings of personal control [29,37] and positive emotionality [38]). Unlike collective narcissists, these individuals are less interested in joining a strong group in order to help them cope with potential psychological shortcomings [30]. Instead, persons who score high on secure national identification search for an opportunity to constructively develop or strengthen the image of their national ingroup by working on behalf of it (self in the service of the group [36]). Consequently, they are not hostile against out-group members, do not endorse conspiratorial explanations [16], and have positive attitudes towards both ingroup and out-group members [16,29,30]. Recent research also found that those securely identified with their nation are more trusting, open-minded, and ready to respect the views and opinions of others, even if they disagree with them [39]. All these findings suggest that secure identifiers should behave differently than collective narcissists do. They should be less likely to believe in conspiracy theories and consequently less likely to refuse COVID-19 vaccination.

Likewise, we would expect similar dynamics among those individuals who are not only positively attached to their nation, but also feel a connection with people far beyond their current geographical location, labeled as an identification with all humanity (IWAH [31,42]). Referring to all “humanity” as an ingroup equals moving beyond parochial interests towards solidarity with and care for all humans [31]. This type of identity is present among psychologically mature individuals who have developed “deep feelings of identification, sympathy, and affection for human beings in general” [43] (p. 138), and is linked to engaging in activities that express positive attitudes towards all humankind [31,44]. IWAH going hand in hand with psychological security was documented in studies on attachment styles. High scores on the IWAH scale were positively correlated with a secure attachment style, and negatively with fearful attachment style [34]. Furthermore, IWAH was negatively related to the need for approval [31], self-centeredness [45], and “dark triad” personality traits [31,46]. When it comes to relationships between IWAH and group-related outcomes, McFarland [47] and Hamer with colleagues [48] presented evidence for its negative relation with ethnocentrism. Likewise, IWAH was found to negatively predict prejudice and intergroup hostility [49], but it was positively related to forgiveness of former national enemies [34,50], and intergroup empathy and helping [51].

### 1.3. Overview of Current Research

We examined our predictions in two cross-sectional studies conducted in Poland, where uptake of COVID-19 vaccines among citizens is lower than the EU average (i.e., only 54% of Poles were fully vaccinated in December 2021 [52]). We assumed that higher national narcissism (rather than national identification, Studies 1 and 2, or identification with all humanity, Study 2) would be associated with lower willingness to vaccinate against COVID-19 (H1), and that that this effect would be mediated by COVID-19 vaccine conspiracy beliefs (H2). We expected the relationship between national narcissism and COVID-19 vaccination attitudes to be especially strong when controlled for the shared variance between national narcissism and national identification. For this reason, in Studies 1 and 2, we first report zero-order correlations. Next, we report the effects of national narcissism and national identification on COVID-19 vaccine conspiracy beliefs and the willingness to vaccinate against COVID-19 while considering both national narcissism and national identification in one model (i.e., controlling for their shared variance). In both studies, we also controlled for basic demographics (i.e., age, gender, education, and size of place of residence); in Study 2, we additionally accounted for the effect of identification with all humanity. Our research included at least 400 participants, which gave us a power of 0.80 for detecting even small associations between studied variables (for r = 0.14; [53]; G*Power yields a target of 395 participants). Research procedures were reviewed and approved by an institutional ethics committee.

## 2. Study 1

In Study 1, we examined the relationships between different forms of national identity and attitudes toward COVID-19 vaccination. To this end, we analyzed data from a nationwide study that assessed national narcissism, national identification, COVID-19 vaccine conspiracy beliefs, willingness to vaccinate against COVID-19, and demographic factors, among other variables. We assumed that national narcissism (but not national identification) would be a negative predictor of the willingness to vaccinate against COVID-19, and that this relationship would be accounted for by COVID-19 vaccine conspiracy beliefs.

### 2.1. Methods

#### 2.1.1. Participants and Procedure

This study involved a nationwide sample of Polish adults. The survey was completed at the beginning of COVID-19 pandemic (March 2020). The sample consisted of 432 participants (42% female, coded as 1), aged between 18 and 84 (*M* = 48.18; *SD* = 16.34). Data were collected via an Internet questionnaire (CAWI) by a Polish online survey research platform (Pollster Institute, Warsaw, Poland) that also conducts academic studies [11]. Respondents were randomly selected from a nationally representative online panel. Sample characteristics can be considered as representative of Polish adults regarding their age, gender, and place of residence.

#### 2.1.2. Measures

**National narcissism** was measured in relation to the national group with a short 5-item version of the Collective Narcissism Scale [20]. Participants indicated to what extent they agreed with statements such as “I will never be satisfied until Poles get all they deserve” or “If Poles had a major say in the world, the world would be a much better place”, on a scale from 1 = *definitely disagree* to 5 = *definitely agree* (α = 0.92, *M* = 2.86, *SD* = 1.09). Higher mean scores indicated higher national narcissism.

**National identification** was measured using a 12-item social identity scale by Cameron [54] (for the Polish adaptation, see Bilewicz and Wójcik [55]). Sample items were: “Being a Pole is an important part of my self-image”; “In general, I’m glad to be a Pole”; “I have a lot in common with other Poles”. Participants responded on a scale from 1 = *definitely disagree* to 5 = *definitely agree* (α = 0.90, *M* = 3.59, *SD* = 0.76). Higher mean scores indicated higher national identification.

**COVID-19 vaccine conspiracy beliefs** were measured with 5-item scale on the basis of the scale of vaccination conspiracy beliefs previously used by Jolley and Douglas [56]: “COVID-19 vaccines are harmful, and this fact is covered up”; “Tiny devices are placed in COVID-19 vaccines to track people”; “Pharmaceutical companies, scientists, and academics work together to cover up the dangers of COVID-19 vaccines”; “COVID-19 vaccines will cause autism”; “The COVID-19 vaccine allows the government to monitor the elderly through the implantation of tiny tracking devices”. Participants responded on a scale from 1 = *definitely disagree* to 5 = *definitely agree* (α = 0.93, *M* = 2.52, *SD* = 1.15). Higher mean score indicated higher COVID-19 vaccine conspiracy beliefs.

**Willingness to vaccinate against COVID-19** was measured using one dichotomous item: “Will you vaccinate against COVID-19 if there is a vaccine available in Poland?” The answers were recoded as “0” (*No*) and “1” (*Yes*). Of participants, 51% (*n* = 220) declared willingness to vaccinate against COVID-19.

**Control variables** involved gender (0 = male, 1 = female), age, education (1 = below secondary, 2 = secondary, 3 = higher), and size of place of residence (1 = village, 2 = town with fewer than 100,000 residents, 3 = town of between 100,001 and 499,999 residents, 4 = town with 500,000 residents or more).

### 2.2. Results

#### 2.2.1. Zero-Order Correlations

We first computed correlations between variables (Table 1). National narcissism and national identification were significantly positively correlated with each other. As predicted, national narcissism was positively related to COVID-19 vaccine conspiracy beliefs and negatively to willingness to vaccinate against COVID-19. However, national identification was not significantly related to either COVID-19 vaccine conspiracy beliefs or willingness to vaccinate against COVID-19. As expected, COVID-19 vaccine conspiracy beliefs were negatively related to the willingness to vaccinate against COVID-19.

#### 2.2.2. Regression Analysis

We were primarily interested in examining whether national narcissism was positively associated with COVID-19 vaccine conspiracy beliefs and negatively with willingness to vaccinate against COVID-19. Thus, we examined the regression models with national narcissism and national identification as predictors of COVID-19 vaccine conspiracy beliefs (DV1, i.e., first dependent variable) and willingness to vaccinate against COVID-19 (DV2), controlling for demographic variables.

**COVID-19 vaccine conspiracy beliefs (DV1).** With both identity variables included in the equation, we found that national narcissism positively predicted COVID-19 vaccine conspiracy beliefs, whereas this association was negative for national identification. As illustrated in Table 2, these effects remained significant after controlling for demographic variables.

**Willingness to vaccinate against COVID-19 (DV2).** To test the hypothesis that national narcissism (but not national identification) would predict lower willingness to vaccinate against COVID-19, we conducted hierarchical binominal regression analysis. Analysis controlled for demographics. In Step 1, we introduced both forms of national identity. As predicted, national narcissism negatively, and national identification positively predicted willingness to vaccinate. In Step 2, we introduced COVID-19 vaccine conspiracy beliefs which were negatively associated with willingness to vaccinate. After adding COVID-19 vaccine conspiracy beliefs, the effects of national narcissism and national identification on willingness to vaccinate were no longer significant. This pattern of results did not change after controlling for demographics in Step 3, with age being a significant (and positive) predictor of willingness to vaccinate against COVID-19 (Table 3).

#### 2.2.3. Mediation Analyses

To perform a full test of our hypotheses, we conducted mediation analysis using Process Software 4.0 (Process Software, Framingham, MA, USA) ([57]; Model 4). We tested whether COVID-19 vaccine conspiracy beliefs mediated the path between national narcissism (vs. national identification) and willingness to vaccinate, controlling for the effects of demographics. Results are presented in Figure 1.

The negative indirect effect of national narcissism on the willingness to vaccinate via COVID-19 vaccine conspiracy beliefs was significant with a bootstrapped (5000 resamples) 95% bias-corrected confidence interval, *IE* = −0.67, *SE* = 0.13, 95% *CI* [−0.96, −0.47].

The positive indirect effect of national identification on willingness to vaccinate via COVID-19 vaccine conspiracy beliefs was also significant, *IE* = 0.52, *SE* = 0.15, 95% *CI* [0.26, 0.86].

### 2.3. Discussion

Study 1 showed that national narcissism was significantly associated with higher COVID-19 vaccine conspiracy beliefs and lower willingness to vaccinate against COVID-19. These effects were demonstrated over and above the effects of demographic variables. After controlling for the shared variance between both types of national identity, secure national identification predicted lower COVID-19 vaccine conspiracy beliefs and higher willingness to vaccinate against COVID-19, confirming Hypotheses 1 and 2. Most importantly, the relationship between national narcissism (and national identification) and the willingness to vaccinate against COVID-19 was mediated by COVID-19 vaccine conspiracy beliefs. Thus, the study provided the first evidence that national narcissism and secure national identification may have opposite relationships with COVID-19 vaccine hesitancy.

## 3. Study 2

The purpose of Study 2 was twofold: we aimed to replicate the effects observed in Study 1 and to extend this model to include potential effects of identification with all humanity (IWAH). As previously mentioned, IWAH served as a significant predictor of health behaviors [23,24]. It was also related to a sense of individual security [50] and positive intergroup relations [31,34,50]. For these reasons, we decided to control for this broad identification in Study 2.

### 3.1. Methods

#### 3.1.1. Participants and Procedure

Data for Study 2 were obtained through an online survey (CAWI) conducted in Poland in early December 2020. Respondents were randomly selected from a nationally representative online panel. Data collection was performed online by a survey research company (Ogólnopolski panel badawczy Ariadna, Warsaw, Poland) also conducting academic investigations [13]. The final sample for this study included 807 Polish participants (51% female, coded as 1), aged between 18 and 85 (*M* = 45.68, *SD* = 15.49); it was representative of Polish adults in terms of gender, age, and size of place of residence.

#### 3.1.2. Measures

**National narcissism** was measured with the same 5-item version of the Collective Narcissism Scale [20] as in Study 1, again assessed in relation to the national group (α = 0.95, *M* = 2.90, *SD* = 1.13).

**National identification** was measured with a shortened 3-item Social Identification Scale ([54]; see also [41]). The three items read: “I have a lot in common with other Poles”, “In general, being a Pole is an important part of my self-image”, “Generally, I feel good when I think about myself as a Pole” (α = 0.93, *M* = 3.62, *SD* = 1.01).

**Identification with all humanity** was measured with a 9-item IWAH scale [42] in the Polish adaptation by Hamer and colleagues [58], with such items as “How close do you feel to each of the following groups…” “people all over the world”; “How much would you say you care (feel upset, want to help) when bad things happen to…?”; “all humans everywhere”). All items used 5-point response scale, but the anchors differed on the basis of question wording (i.e., 1 = *not at all* to 5 = *very close*; 1 = *almost never* to 5 = *very often*; α = 0.94, *M* = 2.91, *SD* = 0.81). Higher mean scores indicated higher identification with all humanity.

**COVID-19 vaccine conspiracy beliefs** were measured with the same 5-item scale as in Study 1 (α = 0.92, *M* = 2.60, *SD* = 1.11).

**Willingness to vaccinate against COVID-19** was measured using one item: “Do you plan to vaccinate against COVID-19 if there is a safe, tested vaccine available?”, with five answer options (1 = *definitely no* to 5 = *definitely yes*; *M* = 2.94, *SD* = 1.43).

**Control variables**. Following Study 1, analyses adjusted for gender (0 = male, 1 = female), age, education (1 = primary, 2 = vocational, 3 = high school, 4 = postsecondary, 5 = bachelor degree, 6 = master degree, 7 = postgraduate degree), and size of place of residence (1 = village, 2 = town up to 20,000 residents, 3 = town between 20,001 and 100,000 residents, 4 = town between 100,001 and 500,000 residents, 5 = town above 500,000 residents).

### 3.2. Results

#### 3.2.1. Zero-Order Correlations

We first computed correlations between continuous variables (see Table 4). National narcissism and national identification were significantly positively correlated with each other (Table 4). Predictably, national narcissism was positively related to COVID-19 vaccine conspiracy beliefs. This time, however, we did not find a significant relationship between national narcissism and willingness to vaccinate against COVID-19. National identification was not significantly related to COVID-19 vaccine conspiracy beliefs. However, we found a significant positive association between national identification and willingness to vaccinate against COVID-19. Identification with all humanity was not related to COVID-19 vaccine conspiracy beliefs, but it was significantly related to willingness to vaccinate against COVID-19. IWAH was positively related to both forms of national identity. Conspiracy beliefs about COVID-19 vaccines were negatively related to willingness to vaccinate, as in Study 1.

#### 3.2.2. Regression Analysis

As in Study 1, we examined the regression models with national narcissism and national identification as predictors of COVID-19 vaccine conspiracy beliefs (DV1) and willingness to vaccinate against COVID-19 (DV2), controlling for demographic variables and identification with all humanity.

**COVID-19 vaccine conspiracy beliefs (DV1).** As predicted, when we simultaneously entered national narcissism and national identity into our regression equation, both emerged as significant predictors of COVID-19 vaccine conspiracy beliefs. As expected, their relationships with conspiracy beliefs were in opposite directions. These associations remained statistically significant after controlling for identification with all humanity and demographics (Table 5).

**Willingness to vaccinate against COVID-19 (DV2).** To test the hypothesis that national narcissism (but not national identification) would predict lower willingness to vaccinate against COVID-19, we conducted additional hierarchical regression analysis (Table 6). In Step 1, national narcissism negatively predicted willingness to vaccinate. On the other hand, national identification had a positive effect on this outcome. In Step 2, identification with all humanity was positively related to willingness to vaccinate. The effects of national narcissism and national identification remained statistically significant, albeit weaker. In Step 3, we entered COVID-19 vaccine conspiracy beliefs into the equation. The effect of this variable on willingness to vaccinate was negative. After adding COVID-19 vaccine conspiracy beliefs, we found significant and positive effects of national narcissism and identification with all humanity on willingness to vaccinate. The effect of national identification on willingness to vaccinate was no longer statistically significant. This pattern of results did not change after controlling for demographic variables in Step 4 (Figure 2), with age being a significant (and positive) predictor of willingness to vaccinate against COVID-19 (Table 6). We also found that males scored significantly higher on willingness to vaccinate than females did.

#### 3.2.3. Mediation Analyses

Mediation analysis (Figure 2) using Process Software 4.0 ([57]; model 4) followed. Similar to Study 1, the statistical significance of the indirect effects was established on the basis of 95% bias-corrected bootstrap (5000 re-samples) confidence intervals. In accordance with our expectations, there was a negative indirect effect of national narcissism on willingness to vaccinate via COVID-19 vaccine conspiracy beliefs, *IE* = −0.20, *SE* = 0.03, 95% *CI* [−0.26, −0.15]. As predicted, the indirect effect of national identification on willingness to vaccinate through COVID-19 vaccine conspiracy beliefs was positive and significant, *IE* = 0.08, *SE* = 0.03, 95% *CI* [0.02, 0.04]. By contrast, the positive effect of identification with all humanity on the DV was not mediated by COVID-19 conspiracy beliefs, *IE* = −0.01, *SE* = 0.03, 95% *CI* [−0.08, 0.05].

### 3.3. Discussion

Findings of Study 2 replicated the pattern of results obtained in Study 1 and showed that national narcissism (but not secure national identification) was associated with greater endorsement of COVID-19 vaccine conspiracy beliefs and in turn greater unwillingness to vaccinate against COVID-19. As in Study 1, we found that secure identifiers rejected conspiracy beliefs and thus exhibited a greater intention to vaccinate against COVID-19. Study 2 additionally revealed that identification with all humanity did not significantly predict COVID-19 vaccine conspiracy beliefs. However, and more importantly, in line with previous research findings demonstrating its connection to health behaviors during pandemics [23], our analyses showed that IWAH was also positively associated with willingness to vaccinate against COVID-19.

## 4. General Discussion

Since the effectiveness of a vaccination campaign is largely dependent on wide vaccine uptake, this research program aimed to better understand psychological factors that could be associated with (un)willingness to get vaccinated against COVID-19. We conducted our investigations in Poland, a country that belongs to the group of Central and Eastern European nations with the lowest vaccination acceptability [59]. In two studies conducted with nationwide samples, we examined the links between different types of social identities (i.e., national narcissism, national identification, identification with all humanity), COVID-19 vaccine conspiracy beliefs and (un)willingness to vaccinate against COVID-19.

In Studies 1 and 2, we established that national narcissism emerged as a robust predictor of COVID-19 vaccine conspiracy beliefs, which in turn significantly predicted lower willingness to get vaccinated. Results of both studies also showed that, when we controlled for the variance shared between national narcissism and national identification, the latter became a significant and positive predictor of willingness to vaccinate against COVID-19. In other words, secure identifiers rejected conspiracy theories, thus being more willing to get COVID vaccines. Thus, it seems that only defensive type of national identity, which is associated with sensitivity to intergroup threats [19], fosters conviction that there is an evil outgroup that secretly uses COVID-19 vaccines for their own nefarious purposes. In such a way, our investigations corroborated and extended previous findings linking national narcissism to vaccination conspiracy beliefs [22] in a different context. We showed that national narcissists are not only more prone to believe in conspiracy theories, but also that due to these beliefs, they are less willing to vaccinate against COVID-19.

Importantly, in Study 2, we additionally demonstrated that not only more specific (i.e., national), but also broad (i.e., identification with all humanity) social identities could be related to COVID-19 vaccine acceptance. Results revealed that those individuals who scored higher on identification with all humanity scale (i.e., ascended from parochial interests to solidarity with and care for all humans; [34]) were more willing to vaccinate against COVID-19. However, IWAH and COVID-19 vaccine conspiracy beliefs were not associated with each other. The mechanism operating behind the link between IWAH and vaccination attitudes may be more complex. The results of Barragan and colleagues’ [23] study documented that identification with all humanity can be considered “the most consistent and consequential predictor of individuals’ cooperative health behavior and helpful responding” during the COVID-19 pandemic [23] (p. 1). Zagefka’s [60] research on prosociality during COVID-19 showed that a focus on global solidarity (also in the form of identification with all humans) enhanced helping within- and across-group boundaries. Thus, examinations of global human identity and COVID-19 attitudes and behaviors, including our study, showed that such a broad identification may be especially potent in promoting prosocial attitudes and behaviors that contribute to protecting fellow humans during a global pandemic [61]. Therefore, we surmise that the positive relationship between identification with all humanity and willingness to vaccinate against COVID-19 may be explained to an extent by prosocial propensities that should be particularly salient in times of a global emergency, in times of a threat to all humans. The credence of this explanation should be tested in future studies.

Additional investigations could also attempt to establish the causal links of observed relationships, for example, by experimentally manipulating the levels of national narcissism and COVID-19 vaccine conspiracy beliefs. It is reasonable to expect that boosting (vs. attenuating) national narcissism may result in higher (vs. lower) levels of conspiracy beliefs. On the other hand, we cannot exclude the possibility that boosting (vs. attenuating) COVID-19 conspiracy beliefs may change the levels of national narcissism [18]. Another potentially fertile ground for future research would be to replicate these results in other countries and cultures [62], as the link between COVID-19 vaccine conspiracy beliefs and vaccine hesitancy was established, for example, in the United Kingdom [63], United States [64], Jordan, Kuwait, and other Arab countries [65].

## 5. Conclusions

Overall, our research complements the broader literature on the role of social identities in adopting conspiracy theories and forming anti-science attitudes. It shows that, although some types of social identities (e.g., national narcissism) are positively linked to rejecting science, others (e.g., secure national identification or identification with all humanity) have the potential to strengthen acceptance of scientific claims, leading to positive health and prosocial behaviors. Thus, more empirical efforts should focus on finding an effective way to boost secure national identification and identification with all humanity, as both these factors seem to be promising for reducing COVID-19 and other vaccine hesitancy.

## Figures and Tables

**Figure 1 vaccines-10-00268-f001:**
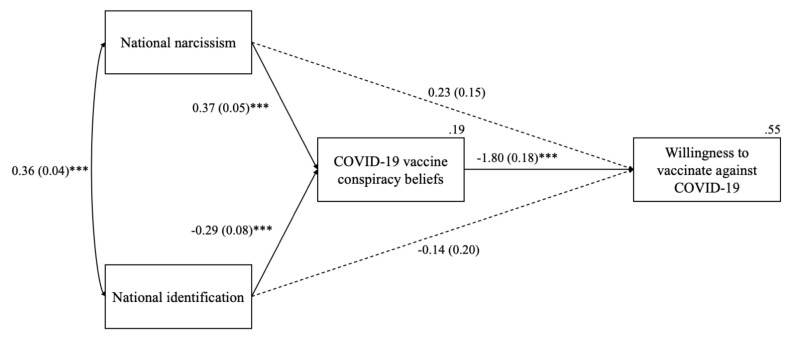
Indirect effects of national narcissism and national identification on willingness to vaccinate via COVID-19 vaccine conspiracy beliefs (Study 1). *Note*: Entries are unstandardized regression coefficients with standard errors in parentheses. Solid lines represent significant effects and dashed lines depict nonsignificant effects. Gender, age, education, and size of place of residence were controlled for. Total effects of national narcissism and national identification were *B* = −0.44, *SE* = 0.18, *p* = 0.015 and *B* = 0.38, *SE* = 0.25, *p* = 0.135, respectively. *** *p* < 0.001.

**Figure 2 vaccines-10-00268-f002:**
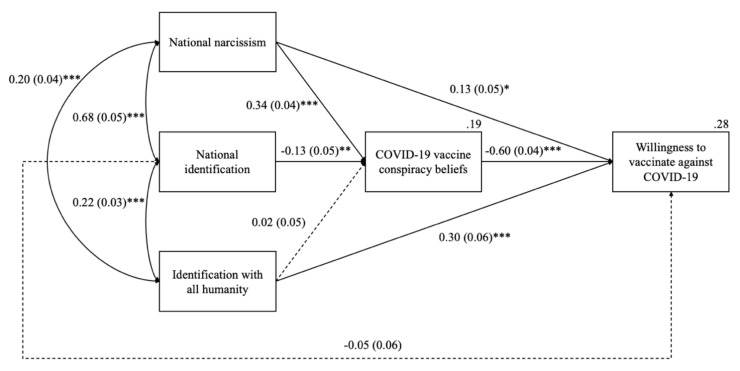
Indirect effects of national narcissism, national identification, and identification with all humanity on willingness to vaccinate via COVID-19 vaccine conspiracy beliefs (Study 2). *Note:* Entries are unstandardized regression coefficients with standard errors in parentheses. Solid lines represent significant effects and dashed lines depict nonsignificant effects. Gender, age, education, and size of place of residence were controlled for. The total effects of national narcissism, national identification and identification with all humanity were *B* = −0.07, *SE* = 0.06, *p* = 0.187, *B* = 0.02, *SE* = 0.06, *p* = 0.720, and *B* = 0.29, *SE* = 0.06, *p* < 0.001, respectively. * *p* < 0.05 ** *p* < 0.01. *** *p* < 0.001.

**Table 1 vaccines-10-00268-t001:** Zero-order correlations (Study 1).

Measure	1	2	3	4
1. National narcissism	-	0.44 ***	0.31 ***	−0.16 ***
2. National identification		-	−0.08	0.08
3. COVID-19 vaccine conspiracy beliefs			-	−0.65 ***
4. Willingness to vaccinate against COVID-19				-

*** *p* < 0.001.

**Table 2 vaccines-10-00268-t002:** National narcissism and national identification predicting COVID-19 vaccine conspiracy beliefs, controlling for demographics (Study 1).

Variable		Step 1			Step 2	
	***B* (*SE*)**	**β**	** *p* **	***B* (*SE*)**	**β**	** *p* **
National narcissism	0.45 (0.05)	0.42	<0.001	0.37 (0.05)	0.35	<0.001
National identification	−0.39 (0.07)	−0.26	<0.001	−0.29 (0.08)	−0.19	<0.001
Gender				0.23 (0.10)	0.10	0.03
Age				−0.008 (0.003)	−0.12	0.012
Education				−0.16 (0.06)	−0.11	0.011
Place of residence size				−0.09 (0.05)	−0.08	0.065
*F*		37.72	<0.001		17.12	<0.001
*R* ^2^		0.15			0.20	
Δ*R*^2^		0.15			0.05	
Δ*F*		37.72	<0.001		5.95	<0.001

**Table 3 vaccines-10-00268-t003:** National narcissism, national identification, and COVID-19 vaccine conspiracy beliefs predicting willingness to vaccinate against COVID-19, controlling for demographics (Study 1).

Variable		Step 1			Step 2			Step 3	
	*B* (*SE*)	OR	*p*	*B* (*SE*)	OR	*p*	*B* (*SE*)	OR	*p*
National narcissism	−0.47 (0.11)	0.63	<0.001	0.17 (0.15)	1.18	0.261	0.23 (0.15)	1.26	0.134
National identification	0.50 (0.15)	1.65	<0.001	0.001 (0.19)	1.00	0.995	−0.14 (0.21)	0.87	0.503
Vaccine conspiracy beliefs				−1.80 (0.17)	0.17	<0.001	−1.81 (0.18)	0.17	<0.001
Gender							−0.38 (0.27)	0.68	0.152
Age							0.02 (0.009)	1.02	0.027
Education							−0.14 (0.17)	0.87	0.402
Place of residence size							−0.01 (0.13)	0.99	0.923
Nagelkerke’s *R*^2^		0.07						0.55	
2 log-likelihood		575.41						369.35	

**Table 4 vaccines-10-00268-t004:** Zero-Order Correlations (Study 2).

Measure	1	2	3	4	5
1. National narcissism	-	0.60 ***	0.31 ***	−0.03	0.21 ***
2. National identification		-	−0.05	0.07 *	0.27 ***
3. COVID-19 vaccine conspiracy beliefs			-	−0.47 ***	0.04
4. Willingness to vaccinate against COVID-19				-	0.18 ***
5. Identification with all humanity					-

* *p* < 0.05. *** *p* < 0.001.

**Table 5 vaccines-10-00268-t005:** National narcissism and national identification predicting COVID-19 vaccine conspiracy beliefs, controlling for demographic variables and identification with all humanity (Study 2).

Variable		Step 1			Step 2			Step 3	
	*B* (*SE*)	β	*p*	*B* (*SE*)	β	*p*	*B* (*SE*)	β	*p*
National narcissism	0.42 (0.04)	0.43	<0.001	0.42 (0.04)	0.43	<0.001	0.34 (0.04)	0.35	<0.001
National identification	−0.22 (0.05)	−0.20	<0.001	−0.22 (0.05)	−0.20	<0.001	−0.13 (0.05)	−0.12	0.006
Identification with all humanity				−0.005 (0.05)	−0.003	0.92	0.02 (0.05)	0.02	0.642
Gender							0.13 (.07)	0.06	0.068
Age							−0.02 (0.002)	−0.224	<0.001
Education							−0.05 (0.02)	−0.077	0.019
Residence place size							−0.06 (0.03)	−0.08	0.015
*F*		55.23	<0.001		36.78	<0.001		26.16	<0.001
*R* ^2^		0.12			0.12			0.19	
Δ*R*^2^		0.12			0.00			0.07	
Δ*F*		55.23	<0.001		0.01	0.92		16.11	<0.001

**Table 6 vaccines-10-00268-t006:** National narcissism, national identification, identification with all humanity, and COVID-19 vaccine conspiracy beliefs predicting willingness to vaccinate against COVID-19, controlling for demographics (Study 2).

Variable		Step 1			Step 2			Step 3			Step 4	
	*B* (*SE*)	β	*p*	*B* (*SE*)	β	*p*	*B* (*SE*)	β	*p*	*B* (*SE*)	β	*p*
National identification	0.19 (0.06)	0.14	0.002	0.14 (0.06)	0.10	0.026	−0.004 (0.06)	−0.003	0.941	−0.05 (0.06)	−0.04	0.342
Collective narcissism	−0.14 (0.06)	−0.11	0.013	−0.16 (0.06)	−0.12	0.004	0.11 (0.05)	0.09	0.027	0.13 (0.05)	0.10	0.012
Identification with all humanity				0.32 (0.06)	0.18	<0.001	0.31 (0.06)	0.18	<0.001	0.30 (0.06)	0.17	<0.001
COVID-19 vaccine conspiracy beliefs							−0.64 (0.04)	−0.50	<0.001	−0.60 (0.04)	−0.46	<0.001
Gender										−0.25 (0.09)	−0.09	0.004
Age										0.01 (0.003)	0.12	<0.001
Education										−0.003 (0.03)	−0.003	0.924
Place of residence size										0.01 (0.03)	0.01	0.727
*F*		5.15	0.006		11.78	<0.001		70.62			38.95	<0.001
*R* ^2^		0.013			0.042			0.26			0.28	
Δ*R*^2^		0.013			0.03			0.22			0.02	
Δ*F*		5.15	0.006		24.74	<0.001		236.76	<0.001		5.65	<0.001

## Data Availability

The data that support current findings are openly available in Open Science Framework depository at https://osf.io/y2ekn/ (accessed on 15 January 2022).

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
