# Peer review of "COVID-19: Why Do People Refuse Vaccination? The Role of Social Identities and Conspiracy Beliefs: Evidence from Nationwide Samples of Polish Adults"

_vaccines, 2022, doi:10.3390/vaccines10020268_

Round 1

Reviewer 1 Report

In this interesting manuscript, the authors aimed to to evaluate the role of social identities in seizing on COVID-19 vaccine conspiracy beliefs that would lead to lower willingness to vaccinate against COVID-19.

This is an important topic as COVID-19 vaccination is one of the world’s main topics and the main public health concern in this pandemic period. 

Some comments are made below:

  • This study involved a nationwide sample of Polish adults, so I suggest specifying this in the title. 
  • The aim of this study is clear and the introduction is well written. I appreciated the effort that clearly has been made in reviewing the literature, to give the reader a broad and complete context.
  • In the methods section (for both study 1 and 2), I suggest adding details on how the sample was selected and recruited. 
  • The analysis is well conducted and results well written
  • The “General discussion” highlights important considerations that resulted from this study, but it lacks explanations, comparison with studies conducted in other countries  and investigating similar aspects (find references below that you may find useful), as well as deeper insight on your important results.

Eberhardt J, Ling J. Predicting COVID-19 vaccination intention using protection motivation theory and conspiracy beliefs. Vaccine. 2021 Oct 8;39(42):6269-6275. doi: 10.1016/j.vaccine.2021.09.010. Epub 2021 Sep 7. PMID: 34535313; PMCID: PMC8421109.

Del Riccio M, Boccalini S, Rigon L, Biamonte MA, Albora G, Giorgetti D, Bonanni P, Bechini A. Factors Influencing SARS-CoV-2 Vaccine Acceptance and Hesitancy in a Population-Based Sample in Italy. Vaccines (Basel). 2021 Jun 10;9(6):633. doi: 10.3390/vaccines9060633. PMID: 34200656; PMCID: PMC8228124.

Goffe L, Antonopoulou V, Meyer CJ, Graham F, Tang MY, Lecouturier J, Grimani A, Bambra C, Kelly MP, Sniehotta FF. Factors associated with vaccine intention in adults living in England who either did not want or had not yet decided to be vaccinated against COVID-19. Hum Vaccin Immunother. 2021 Dec 17:1-13. doi: 10.1080/21645515.2021.2002084. Epub ahead of print. PMID: 34919492.

Ruiz JB, Bell RA. Predictors of intention to vaccinate against COVID-19: Results of a nationwide survey. Vaccine. 2021 Feb 12;39(7):1080-1086. doi: 10.1016/j.vaccine.2021.01.010. Epub 2021 Jan 9. PMID: 33461833; PMCID: PMC7794597.

Author Response

Reviewer 1:

R1.1. In this interesting manuscript, the authors aimed to evaluate the role of social identities in seizing on COVID-19 vaccine conspiracy beliefs that would lead to lower willingness to vaccinate against COVID-19. This is an important topic as COVID-19 vaccination is one of the world’s main topics and the main public health concern in this pandemic period. 

We thank the Reviewer for their positive comments.

R1.2. This study involved a nationwide sample of Polish adults, so I suggest specifying this in the title. 

Thank you for this comment! We have now changed the title in line with Reviewer’s suggestion: “COVID-19: why do people refuse vaccination? The role of social identities and conspiracy beliefs. Evidence from nationwide samples of Polish adults.”

R1.3. The aim of this study is clear, and the introduction is well written. I appreciated the effort that clearly has been made in reviewing the literature, to give the reader a broad and complete context.

Thank you! We are glad to hear that!

R1.4. In the methods section (for both study 1 and 2), I suggest adding details on how the sample was selected

Our data were collected by professional and accredited web-based research panels that offered greatest assurance of data integrity and accessibility to all demographic strata characteristic of the Polish population.  Both research panels offer nationally representative databases. Participants were randomly selected from these databases and both samples were generally comparable with Polish internet user population characteristics regarding age, gender, place of residence, and education. 

For both Study 1 and Study 2, we added a couple of sentences about how the samples were selected. 

R1.5. The analysis is well conducted and results well written

Thank you for your positive feedback.

R1.6. The “General discussion” highlights important considerations that resulted from this study, but it lacks explanations, comparison with studies conducted in other countries  and investigating similar aspects (find references below that you may find useful), as well as deeper insight on your important results.

Thank you for these useful suggestions. We have now added references to other studies having established the positive link between COVID-19 conspiracy beliefs and COVID-19 vaccination hesitancy in the UK (Eberhardt & Ling, 2021), U.S. (Ruiz & Bell, 2021), Kuwait, Jordan and other Arab countries (Sallam et al., 2021).

Reviewer 2 Report

The category of secure national identification as opposed to 
narcissistic national identification (NNI) is not convincing. Adherents of the latter are expected to admit also to the former, hence the discriminating power is low. This issue has long history; while Fromm in 1973 (cited as 33) refers to group narcissism as harmful,  Kristeva in 1993 (Nations Without Nationalism, NY, Columbia UP) defends 'good narcissism'. It seems also that the terms  NNI and IWAH are value-laden. However I admit that 'NNI' is established in a recent literature, and the present Authors contributed to it.

In line 164 the Authors write "...we do not assume..." I do not know why they need such a declaration. They do not need to assume anything which should be their result. Please correct or explain this statement.

Regretfully, the pool data have been collected in 2010, where 
anti-vaccine behaviors were still weak in Poland. With more recent 
data, one could expect more clear picture. Also, I would suspect that the collecting of data by internet produces its own bias. A comment on this issue is desired.  Yet, in my opinion, the results are still interesting and deserve publication. 

Author Response

Reviewer 2:

R2.1  The category of secure national identification as opposed to narcissistic national identification (NNI) is not convincing. Adherents of the latter are expected to admit also to the former, hence the discriminating power is low. This issue has long history; while Fromm in 1973 (cited as 33) refers to group narcissism as harmful,  Kristeva in 1993 (Nations Without Nationalism, NY, Columbia UP) defends 'good narcissism'. It seems also that the terms  NNI and IWAH are value-laden. However I admit that 'NNI' is established in a recent literature, and the present Authors contributed to it.

Thank you for this comment. Indeed, the differentiation between national narcissism versus secure national identification is established in a recent literature (Golec de Zavala, Cichocka, & Bilewicz, 2013; JOPY; Marchlewska, Cichocka, Jaworska, Golec de Zavala, & Bilewicz, 2020; BJSP). In our manuscript, we refer to two forms of national identity: the defensive national identity, captured by collective narcissism measured with reference to the national group (net of national identification), and the secure national identity, captured by national identification (net of national narcissism). In what follows, we first present definitions of the two types of identity. We also present past theorizing and review available empirical evidence to argue why we believe collective narcissism and ingroup identification can be considered opposite forms of social identity. In line with previous findings, both causes and consequences of national narcissism versus secure national identification are different (for a review see Cichocka, 2016). We elaborated on these issues in the introduction section:

For example, previous studies showed that narcissistic identification stems from a sense of low personal control [37] and low self-esteem [38]. Collective narcissism has been associated with high defensive self-evaluation [39] and high anxious attachment style [36]. Although national narcissism results from the frustration of different needs, it does not necessarily reinforce feelings of personal control or boost self-esteem [40]. In contrast, due to its defensive nature, it rather backfires leading to maladaptive psychological outcomes. For example, it was found to predict negative view of human nature (i.e., social cynicism; [39]), conspiracy beliefs (e.g., [16]), ingroup disloyalty [29], intergroup hostility (e.g., [18,41]), and lower intergroup forgiveness [34]. Bertin and colleagues [17] reported that national narcissism was associated with rejection of climate science, an effect that was mediated by climate change conspiracy beliefs. (…)

We would expect very different dynamics among secure identifiers whose national commitment stems from satisfied psychological needs (e.g., higher feelings of personal control; [29,40]; positive emotionality; [41]). Unlike collective narcissists, these individuals are less interested in joining a strong group in order to help them cope with potential psychological shortcomings [30]. Instead, persons who score high on secure national identification search for an opportunity to develop or strengthen the image of their national ingroup in a constructive way by working on behalf of it (self in the service of the group; [36]). Consequently, they are not hostile against out-group members, do not endorse conspiratorial explanations (e.g., [16]), and have positive attitudes towards in-group and out-group members alike [16,29,30]. Recent research also found that those securely identified with their nation are more trusting, open-minded, and ready to respect the views and opinions of others, even if they disagree with them [39].

R2.2 In line 164 the Authors write "...we do not assume..." I do not know why they need such a declaration. They do not need to assume anything which should be their result. Please correct or explain this statement.

Thank you very much for this comment. We deleted that entire sentence - indeed, it is not needed in the paper.

R2.3 Regretfully, the pool data have been collected in 2010, where 
anti-vaccine behaviors were still weak in Poland. With more recent 
data, one could expect more clear picture. Also, I would suspect that the collecting of data by internet produces its own bias. A comment on this issue is desired.  Yet, in my opinion, the results are still interesting and deserve publication. 

Data for both studies were collected in 2020, thus during the pandemic. Collecting data online has been the safest, and hence prefered,  method of data gathering during pandemic.  Our data were collected by professional and accredited web-based research panels that offered greatest assurance of data integrity and accessibility to all demographic strata characteristic of the Polish population.  

Nevertheless, it has to be recognized that both our samples may reflect a bias toward more highly educated respondents, as it is usually a case with online studies.